# Belief Inflexibility and Cognitive Biases in Anorexia Nervosa—The Role of the Bias against Disconfirmatory Evidence and Its Clinical and Neuropsychological Correlates

**DOI:** 10.3390/jcm12051746

**Published:** 2023-02-22

**Authors:** Elena Tenconi, Valentina Meregalli, Adriana Buffa, Enrico Collantoni, Roberto Cavallaro, Paolo Meneguzzo, Angela Favaro

**Affiliations:** 1Department of Neuroscience, University of Padova, 35128 Padova, Italy; 2Padova Neuroscience Center, University of Padova, 35129 Padova, Italy; 3School of Medicine, Vita-Salute San Raffaele University, 20132 Milan, Italy; 4IRCCS San Raffaele Turro, Scientific Institute Hospital, 20127 Milan, Italy

**Keywords:** anorexia nervosa, cognitive bias, bias against disconfirmatory evidence (BADE), bias against confirmatory evidence (BACE), liberal acceptance, neuropsychology

## Abstract

The aim of this study was to explore, in a sample of patients with a diagnosis of AN, the ability to question their first impression and, in particular, the willingness to integrate their prior ideas and thoughts with additional progressive incoming information. A total of 45 healthy women and 103 patients with a diagnosis of AN, consecutively admitted to the Eating Disorder Padova Hospital–University Unit, underwent a broad clinical and neuropsychological assessment. All participants were administered the Bias Against Disconfirmatory Evidence (BADE) task, which specifically investigates belief integration cognitive bias. Acute AN patients showed a significantly greater bias toward disconfirming their previous judgment, in comparison to healthy women (BADE score, respectively, 2.5 ± 2.0 vs. 3.3 ± 1.6; Mann–Whitney test, *p* = 0.012). A binge-eating/purging subtype of AN individuals, compared to restrictive AN patients and controls, showed greater disconfirmatory bias and also a significant propensity to uncritically accept implausible interpretations (BADE score, respectively, 1.55 ± 1.6 and 2.70 ± 1.97 vs. 3.33 ± 1.63; Kruskal–Wallis test, *p* = 0.002 and liberal acceptance score, respectively, 1.32 ± 0.93 and 0.92 ± 1.21 vs. 0.98 ± 0.75; Kruskal–Wallis test *p* = 0.03). Abstract thinking skills and cognitive flexibility, as well as high central coherence, are neuropsychological aspects positively correlated with cognitive bias, in both patients and controls. Research into belief integration bias in AN population could enable us to shed light on hidden dimensional aspects, facilitating a better understanding of the psychopathology of a disorder that is so complex and difficult to treat.

## 1. Introduction

Anorexia nervosa (AN) is a psychiatric disorder characterized by weight loss, extreme fear of becoming fat, and severe body image distortion [1]. Central to cognitive theories of eating disorders is the hypothesis that beliefs and expectancies referring to both body size and eating are biased in favor of selectively processing information that is related to body weight, dieting, and control of food intake [2]. Other types of biases probably play a role in eating disorders. For example, recent literature shows that the processing of social stimuli is negatively biased in patients with AN [3,4]. Cognitive biases and distortions are hypothesized to contribute to the development and maintenance of the disorder and may interfere with treatment outcomes. A better understanding of information processing biases in AN is crucial in order not only to understand the “core” psychopathological features (including illness denial and resistance to treatment) but also to provide targeted and patient-tailored interventions. The etiopathogenetic model of eating disorders is multifactorial and identifies, among others, adverse early life experiences as those factors negatively affecting, with long-lasting impact, physical and mental health. Several studies endorse childhood adversity and other types of early life adversities as predisposing factors to the development of AN and eating disorders later in life [5]. These experiences potentially impact brain function and structure, with particular involvement of certain brain areas (such as the prefrontal cortex, amygdala, and hippocampus), critical for cognitive and emotional regulation [6]. In line with these considerations, early adverse experiences may negatively affect thinking style, at least in part, leading to both neuropsychological alterations and cognitive biases, besides the formation of eating-disordered typical, rigid, hostile internal dialogue known as the “inner voice of anorexia” later in life [7]”.

From a psychopathological point of view, typical ideas of patients with AN are included among over-valued thoughts [1]. Over-valued thinking is commonly considered in the middle of a continuum between pathological doubts characterizing obsessional thinking and strong or irrefutable certainty pertaining to delusional thoughts or, more simply, is considered similar to obsessive thought but associated with high ego-syntonicity and poor insight. Over-valued ideas can be considered “idealized” values that the individual has developed in such an overriding, rigid, and unquestionable importance, that they totally define the “self” or identity of the individual; these ideas are pursued without adaptation to different circumstances and ignoring the consequences of acting on their value [8]. Although over-valued thought is the more appropriate definition of the pathological thought process that characterizes patients with AN, in some cases patients also display strong irrational beliefs about the effects of food, energy requirements, or body size and shape (see, for example, the cognitive distortion named “thought-shape fusion”) [9,10], which resemble the characteristics of delusional beliefs [11,12]. In recent years, the literature on eating disorders has started to describe examples of these “delusional” ideas [13,14]. In both research papers [15] and case studies [16,17,18] the delusional rigid confidence in their own beliefs of patients affected by AN was found to be similar to that observed in patients with a diagnosis of psychosis. Delusional beliefs were observed in a subgroup of AN patients (about 20–30%), especially those belonging to the restrictive subtype, and appear to be more correlated to the psychopathological severity of the disease (e.g., the “drive for thinness” scale of EDI), than to typical prognostic indicators (i.e., BMI or duration of the disorder) [11,14]. It is possible that in a subgroup of “at-risk” patients or in particularly vulnerable situations [19], over-valued ideas may progress towards delusional thought. This type of thought is characterized by belief rigidity and the inability to doubt the appropriateness of our own ideas to the context, in addition to difficulties in integrating different evidence in reasoning processes. In general, we can define cognitive biases as systematic errors in information processing that automatically affect (i.e., unconsciously) judgments and decisions, de facto altering thinking ability. The landscape of cognitive biases is extremely broad (e.g., implicit bias, priming bias, confirmation bias, affinity bias, belief bias, etc.). These biases may affect human behavior throughout belief formation, reasoning processing, and decision-making. In certain contexts, some of these biases enable faster decisions (e.g., heuristics) and are considered useful and adaptive; nevertheless, in psychiatric conditions, systematic biases may function as illness-maintaining factors and interfere with taking charge and responding to the treatment. Biased cognitive processes are indeed involved in the formation and maintenance of delusional beliefs [20] and they probably play a role in the etiopathogenesis of some psychiatric disorders [21]. “Jumping to conclusions” (JTC) and “Bias against Disconfirmatory Evidence” (BADE) are the most commonly described biases in this context, as both reflect the tendency to interpret reality or to make decisions with certainty based on inadequate and premature evidence; the so-called “data gathering bias” [22]. Moreover, this bias also regards confirmatory reasoning, which entails difficulties in integrating new increasingly contradicting evidence into the reasoning process in order to reconsider a prior idea [21,23]. Some ecological tasks, the “beads task” and its different versions (i.e., “fish task” or “word/survey task”) for JTC and the BADE task for “Bias against Disconfirmatory Evidence” have been developed to measure these abilities. The literature shows that patients with schizophrenia who are delusional at the time of assessment display significantly higher JTC and BADE biases than patients with the same diagnosis without current delusions. In the latter subgroup, the JTC is similar to that of healthy controls [24]. It is noteworthy that these cognitive biases appear to be associated with the presence of delusions across different psychiatric diagnoses (not only schizophrenia), contributing to the maintenance, severity, and persistence of delusional symptoms in psychiatric disorders [25]. Recently, a belief integration bias tendency, assessed by the BADE task, seems to contribute to maintaining conspiracy theory beliefs in the general population [26]. Moreover, with the aim of investigating the relationship between performance at BADE and psychotic experiences in a group of adolescents drawn from the general population, the authors identified hallucinations, but not paranoia, as a predictor of belief inflexibility [27].

To date, very few studies have explored these types of cognitive biases in patients with eating disorders. Although no study has explored the presence of BADE in patients with eating disorders, the JTC bias in AN patients has been investigated by two studies [28,29], which found no evidence in support of the presence of this type of bias in AN patients.

The main aim of the present study was to specifically investigate the presence of biases against disconfirmatory evidence in a sample of patients with AN. We also aim to explore the relationship between BADE and the clinical characteristics of patients, including their neuropsychological functioning. We hypothesize that AN patients have impaired abilities in processing evidence integration and that the degree of this impairment correlates with both clinical severity and neuropsychological profile. Finally, given the importance of improving our knowledge on such a challenging disorder, we are also interested in assessing the prognostic role of these cognitive biases.

## 2. Materials and Methods

A sample of 82 female patients with a current diagnosis of AN (acute AN) and 45 healthy female individuals participated in this study. A group of patients (*n* = 21) with lifetime AN in partial remission (normal weight, but still in treatment for their disorder) (weight-recovered AN) was included to better analyze the effects of malnutrition. Patients were consecutively recruited among all those in outpatient treatment at the Eating Disorder Unit of the University Hospital of Padova, Italy, in the period between April 2015 and January 2018. The task was administered to all patients who underwent neuropsychological assessment in this period according to inclusion/exclusion criteria in this study as previously reported [30] and willingness to participate. The healthy control group was recruited from the community as described in previous studies [31,32,33]. Demographic and clinical characteristics of the samples are reported in detail below (see Section 3 Results).

This study obtained the approval of the Institutional Review Board and all the participants (all the parents, in the case of minors) gave informed written consent for the use of data in an anonymous form, in accordance with the latest revision of the Helsinki Declaration (2013) [34].

Inclusion criteria for patients were: more than 14 years of age, admission to our Eating Disorder Unit, a diagnosis of AN, and written informed consent. Exclusion criteria were: prior or current traumatic brain injury, lifetime history of any neurological, systemic and/or severe psychiatric illness in comorbidity with AN (suicidality, alcohol/substance use, psychotic features). The exclusion criteria for controls, recruited from the general population, were body mass index (BMI) below 18, having a first-degree relative with a lifetime eating disorder, prior or current traumatic brain injury, any neurological, psychiatric, or systemic illness, and use of psychoactive medication.

### 2.1. Clinical Assessment

All participants (patients and controls) underwent a baseline assessment including the eating disorders section of the Structured Clinical Interview for DSM-5 (SCID) [35], and a semi-structured interview about sociodemographic and clinical characteristics. All the included patients have a lifetime diagnosis of AN according to the DSM-5 criteria [1].

All participants completed some self-reported questionnaires, such as the Hopkins Symptoms Check List (H-SCL-90) [36], which investigates the presence and severity of psychiatric symptoms, the Eating Disorders Inventory (EDI) [37], as a measure of eating psychopathology, and the State-Trait Anxiety Inventory form Y (STAI) [38], which measures anxiety in both its components (i.e., state and trait). We tested correlations of the BADE task performance with both depression and obsessive–compulsive symptoms (H-SCL-90), drive for thinness, body dissatisfaction, ineffectiveness and interoceptive awareness (EDI), and anxiety (STAI). The validated Italian versions of the questionnaires showed good reliability in our sample (Cronbach’s alpha greater than 0.85). The Edinburgh Handedness Inventory [39] was administered to assess hand lateralization with a score from −100 (fully left-handed) to +100 (fully right-handed).

### 2.2. Neuropsychological Assessment

All participants, AN patients and controls, were administered a neuropsychological battery covering a broad range of cognitive functions (for a full description see 30 and 31) including, in particular, the BADE task, the Wisconsin Card Sorting Test (WCST), the Rey–Osterrieth complex figure test (ROCFT), and previous IQ measures.

### 2.3. The BADE Task

The task [21] assesses disconfirmatory/confirmatory cognitive biases and the process of evidence integration. The BADE task consists of 30 scenarios, each of which is composed of three successive statements that progressively disambiguate the context, giving an increasingly available amount of information. The first statement is followed by four interpretations that must be rated for plausibility. After the first assertion, participants are given two further statements and are asked to adjust their prior ratings according to the added slices (pieces) of information. Individuals must rate the four types of possible interpretations for plausibility. These are classified as follows: “true” (ultimate solution; on the first statement this interpretation appears to be either less or equally plausible than the “lure” interpretations and although at first it may appear less convincing, it progressively gains plausibility), “absurd” (implausible interpretation throughout all three statements), “neutral lure” and “emotional lure” interpretations (both explanations that appear plausible for the first statement, but the accumulated evidence across the task should disambiguate their poor plausibility). At every additional statement (three-level information) belonging to each scenario, participants are requested to adjust their plausibility judgment of the four types of independent interpretations (i.e., “true”, “absurd”, and “lure”, both neutral and emotional) given with the task, according to additional information available. Scenarios and positions of four independent interpretations are randomly presented. See Table 1 for an example of a representative trial of the Bias Against Disconfirmatory Evidence (BADE) with three progressive statements (identified with the progressive numbers 1, 2, and 3) and the four sentences (true, neutral lure, emotional lure, and absurd). Plausibility rates were expressed by moving the mouse on a continuous scale (0–10 scrollbar) ranging between “poor” (0) and “excellent” (10) judgments. Participants are asked to change their ratings according to further information added by the three additional statements that progressively specify the context (for a full description of the BADE task and its Italian translation see [40]). The main measures of evidence integration abilities assessed by the task are: (1) Bias Against Disconfirmatory Evidence (BADE), which represents the degree of decrease in plausibility from statement one to statement three, for the “lure interpretations” (both neutral and emotional); (2) bias against confirmatory evidence (BACE) which represents the increase in plausibility for the “true interpretations” from statement one to statement three. Higher change in scores shows higher flexibility skills in evidence integrating; (3) the liberal acceptance index which indicates plausibility ratings of “absurd interpretations”.

In Table 2 the computations of the three outcome variables of the BADE task are summarized, following the methodology of [41]. BADE scores are computed by subtracting plausibility ratings after statement three from those after statement one from emotional and neutral lures averaged together. BACE scores are computed by subtracting the ratings after statement three from the first one. Finally, LA scores are computed as the absolute average of absurd interpretations throughout the task.

#### Interpretation of the BADE Task Scores

Higher BADE scores are suggestive of good abilities to integrate disconfirmatory evidence; higher BACE scores are suggestive of good abilities to integrate confirmatory evidence; lower LA scores are suggestive of the higher ability to refuse absurd interpretations.

Therefore, it follows that low BADE scores mean difficulties in integrating disconfirmatory evidence; low BACE scores suggest difficulties in integrating confirmatory evidence, whereas high LA scores are linked to proneness to accept absurd interpretations as true (i.e., overvaluation of what is implausible).

In Section 3, we also reported data following the scoring system of Woodward et al., 2008 [42]. BADE, BACE, and LA biases are considered dependent variables and are calculated as changing scores computed by averaging together the ratings following statements two and three and subtracting this from the ratings following statement one. For the BADE score (Bias Against Disconfirmatory Evidence) we carried out this procedure for “lure” (both emotional and neutral) interpretations, whereas for the BACE score (bias against confirmatory evidence) we carried it out for “true” interpretations. The LA score (liberal acceptance) is delivered by averaging together all the ratings of “absurd” interpretations following the three statements.

### 2.4. Other Main Cognitive Tasks

−The Wisconsin Card Sorting Test (WCST) [43] assesses abstract thinking and the ability to both infer and adapt to external changes, shifting cognitive strategies (i.e., cognitive flexibility). It is widely considered a good task for assessing the integrity of reasoning and executive functioning. We considered as outcome variables the number of achieved categories, the total number of correct answers, the total number of errors, perseverative responses and errors, and the global score (i.e., an overall index of global cognitive efficiency) [31,44].−The Trail Making Test A and B [45] measures attentional speed, sequencing, visual search, and mental flexibility. Part A (TMT-A) assesses motor speed, part B (TMT-B) assesses complex divided attention and set-shifting, and B-A difference (delta trail) gives a measure of cognitive shifting cost and allows us to control for motor impairment.−The Rey–Osterrieth Complex Figure Test (ROCF) [46], gives a measure of visuospatial constructional abilities and visuographic memory, but cognitive planning, organizational strategies, and executive functions are also involved. The task request is to both directly copy and reproduce, after a delay of 3 min, the complex figure. The former assesses perception and visuospatial constructive abilities, and the latter implicit visuospatial memory. As described elsewhere (for a detailed description see [30,31]), this task was used as a measure of central coherence (the ability to put together different details in order to gain the “big picture”) by means of both the Order of Construction Index (the order in which the different global elements were drawn) and the Style Index (indicative of the degree of continuity in the drawing process). We considered the following outcomes of the ROCF: copy and memory accuracy scores, the time needed to complete the figure copy and the figure reproduction after a delay, and the central coherence index (CCI) of the figure copy. The CCI ranges from 0 (indicating weak central coherence) to 2 (indicating high central coherence).−All participants over the age of 20 completed the Brief Intelligence Test (TIB) [47], as a measure of premorbid intellectual ability. The task requires reading 34 irregular words which violate typical stress rules, very similar to the National Adult Reading Test. All patients and controls younger than 21 were administered the Information subtest of the Wechsler Intelligence Scale-IV edition [48] as a measure of premorbid verbal intelligence.

### 2.5. Statistical Analyses

Analyses were implemented with the software Statistical Product and Service Solutions (SPSS), Version 28 (2021). Variables normally distributed were compared by the statistics of *t*-Student for independent samples. Both the non-parametric Mann–Whitney (for comparisons between two groups) and the Kruskal–Wallis (for more than two groups) tests for independent samples were used. Furthermore, we performed an Analysis of Variance (ANOVA) with age and education as covariates, to investigate the differences between groups checking for possible confounding variables. Correlations were tested with Spearman’s rho coefficient. Given the explorative nature of this study, a probability threshold value of 0.05 was assumed for statistical significance. Moreover, in healthy women, the three outcomes of the task were intercorrelated and a correction for multiple comparisons would impair the sensitivity of the analyses.

## 3. Results

Table 3 shows the main characteristics of participants. The three groups did not differ in age and hand lateralization. Significant differences between acute AN patients and healthy women were observed in education (*t* = 5.16; *p* < 0.001), current BMI (*t* = 15.64; *p* < 0.001), and nadir BMI (lowest lifetime BMI) (*t* = 12.84; *p* < 0.001). The weight-recovered AN group significantly differed from the acute AN group only as regards current BMI (*t* = 3.04; *p* = 0.003), and differed from the group of healthy women as regards education (*t* = 4.59; *p* < 0.001) and nadir BMI (*t* = 5.50; *p* < 0.001).

### 3.1. Acute AN, Weight-Recovered AN, Healthy Women in Comparison, and Disconfirmatory/Confirmatory Cognitive Bias

In Figure 1, average scores of BADE, BACE, and LA are reported. Acute AN patients showed significantly lower flexibility in disconfirming their previous judgments in comparison to healthy women (BADE score, respectively: 2.5 ± 2.0 vs. 3.3 ± 1.6; Mann–Whitney Test U, *p* = 0.012). Given the significant differences between patients and healthy controls in relation to education, we proceeded with an ANOVA, which allows us to control for age and education, thus testing the survival of the Mann–Whitney result. ANOVA, with age and education as covariates, F(3123)= 4.30; *p* = 0.040), confirmed the finding. On the contrary, they did not show difficulties in confirming their previous judgment (BACE), nor did they show greater levels of LA than healthy women. The weight-recovered AN group showed slightly better abilities in integrating disconfirming evidence (BADE) in comparison to the acute AN group, but no statistically significant difference emerged in the comparisons either with patients with acute AN or with healthy women. For this reason, we excluded this group from subsequent analyses. For the analyses of the plausibility rate between healthy women and acute AN, please see Appendix A. We also analyzed possible differences in BADE, BACE, and LA scores according to the age of illness onset (we set the cut-off age at 14 years), but we did not find peculiar patterns or significant differences in scores.

### 3.2. Acute AN Restricting (ANR) and AN Binge-Eating/Purging (ANBP) Subtypes and Healthy Women in Comparison and Disconfirmatory/Confirmatory Cognitive Bias

In the comparison between healthy women and the two diagnostic subgroups of AN patients (ANR, *n* = 69 and ANBP subtype, *n* = 14) a significant difference emerged in the BADE score (healthy women: 3.33 ± 1.63; ANR: 2.70 ± 1.97; ANBP: 1.55 ± 1.62; Kruskal–Wallis test, *p* = 0.002) and in the LA score (healthy women: 0.98 ± 0.75; ANR: 0.92 ± 1.21; ANBP: 1.32 ± 0.93; Kruskal–Wallis test, *p* = 0.03), whereas no difference was found for the BACE score (healthy women: 4.94 ± 1.62; ANR: 5.22 ± 2.11; ANBP: 4.40 ± 3.21; Kruskal–Wallis test, *p* = 0.375). For the BADE score, the ANOVA with age and education as covariates confirmed non-parametric results [F(4123) = 18.28; *p* = 0.005].

The analyses of plausibility rates during the task are reported in Figure 2. A comparison between the two diagnostic subgroups (acute ANR and ANBP) showed a significantly lower BADE score (ANR 2.70 ± 1.97 vs. ANBP 1.55 ± 1.62; Mann–Whitney Test U, *p* = 0.016) and a trend for a higher LA score (ANR 0.92 ± 1.21 vs. ANBP 1.32 ± 0.93; Mann–Whitney Test U, *p* = 0.050) in the ANBP group. The ANBP subtype group also showed lower plausibility ratings at second statement (*p* = 0.043) and at third statement (*p* = 0.006) of the “absurd” sentences, and at third statement (*p* = 0.004) of the “neutral lure” sentences, whereas, at the “emotional lure” sentences, both subtypes of patients appear more insensitive to emotional pitfalls (Figure 2). Healthy women showed greater sensitivity to emotional lure in the first (*p* = 0.031) and second (*p* = 0.006) statements, but finally they appear to recognize the traps (at the third sentence) (Figure 2). Healthy women appeared significantly better than patients at recognizing “true sentences” as true, already from the first statement (*p* = 0.034). The result on neutral and emotional lure sentences, where controls demonstrate significantly more flexible thinking and greater ability to change their minds than patients, is interesting. Patients struggle to integrate disconfirming information and appear insensitive to emotional pitfalls.

### 3.3. BADE Task Performance and Clinical and Psychopathological Variables: Correlational Analyses

Table 4 shows the correlation indices between the BADE task and clinical characteristics in both controls and patients.

In healthy women, the BADE score was significantly inversely correlated with the BACE score (rho = −0.60; *p* < 0.001), but not with the LA score (rho = 0.18; *p* = 0.250). The BACE score was inversely correlated with the LA score (rho = −0.53; *p* < 0.001). In patients with acute AN, the BADE score was not correlated with the BACE and LA scores (rho = −0.17; *p* = 0.13; rho = −0.07; *p* = 0.51), but the BACE score was inversely correlated with the LA score (rho = −0.54; *p* < 0.001). Table 4 shows correlations between the BADE task scores and clinical variables. In healthy women, task performance significantly improved with age, whereas no such relationship was observed in AN patients.

No significant correlations emerged in healthy women or AN patients between BADE, BACE, and LA scores and psychopathology (i.e., anxiety, depression, obsessiveness, …), and clinical characteristics (such as BMI, nadir BMI, age of onset, illness duration).

### 3.4. BADE Task Performance and Neuropsychological Functioning: Correlational Analyses

In healthy women, the BADE score was significantly correlated both with the numbers of correct categories achieved on the WCST (rho = 0.54; *p* < 0.01) and with the copy CCI on the ROCFT (rho = 0.57; *p* < 0.01). In acute AN patients, the BADE score was correlated with the copy CCI on the ROCFT (rho = 0.21; *p* = 0.07) and the TMT-A (rho = 0.21; *p* = 0.06), and both were at trend levels. The BACE score showed significant correlations with the delta trail score on the TMT (rho = 0.28; *p* < 0.01) and a negative correlation with the Memory with Interference at 30” task (rho= −0.22; *p* < 0.05). Lastly, the LA score showed a significant negative correlation with the visuospatial memory score on the ROCFT (rho = −0.29; *p* < 0.01) and a positive correlation with the Memory with Interference task (rho = 0.23; *p* < 0.05).

No other significant correlations emerged in healthy women or AN patients between BADE, BACE, and LA scores and cognitive variables.

## 4. Discussion

The interest on possible overlaps between AN and schizophrenia is a topic that has attracted interest since the earliest formulations of AN [49], given that besides early risk factors [50], the two disorders share some common etiopathogenetic, clinical, neuropsychological, emotional, and interpersonal characteristics [11,51]. Bruch (1973) distinguished primary AN characterized by delusional thinking from atypical non-delusional AN: nonetheless, current diagnostic criteria for AN only include a persistent failure to recognize the severity of the low weight or the obstinate refusal to gain weight [1]. Psychotic experiences seem to contribute as early risk factors to the development of AN and other eating disorders [52]. To date, very little research has been conducted investigating delusional thinking and reasoning bias in clinical populations other than psychoses and delusional patients, and the few existing studies in the eating disorder population focused particularly on the Jumping to Conclusions (JTC) paradigm (along with attributional bias), but the lack of significant findings induced authors to consider the poor insight characterizing AN as different in nature from that observed in schizophrenia [28,29]. In our study, we investigated whether patients affected by AN, compared with a group of healthy women, presented a specific cognitive bias, already explored in schizophrenia and psychoses, namely Bias Against Disconfirmatory Evidence (BADE) and, if so, whether such bias was correlated with demographic, clinical, and/or neuropsychological variables. This is the first study, to our knowledge, addressing evidence integration bias in AN. The main result of our study was that acute AN patients show difficulties in disconfirming their own prior judgments, similar to what is observed in the schizophrenia population, especially in the presence of delusions [24]. Differently from schizophrenia, we failed to find, in acute AN, a bias for confirmatory belief integration (BACE) and high liberal acceptance of absurd explanations (LA). Poor flexibility in disconfirming previous ideas in AN appears to be quite independent of both clinical and psychopathological aspects, in particular body weight, nadir BMI, age of onset, and illness duration. In line with this, weight-restored AN patients investigated in our study showed an overlapping pattern of disconfirmatory/confirmatory integration, just as acute AN individuals, suggesting that the difficulty in integrating disconfirmatory evidence into their ideas, is a relatively stable process, independent of illness severity.

Cognitive biases in AN is a field of increasing interest to clinicians and researchers because patients with AN show a typical clinical presentation that is dominated by specific weight and shape-related distorted cognitions and beliefs, which are usually considered as over-valued ideas [8]. This appears linked to the fact that patients held some awareness that their thoughts and ideas are not objectively true. Nevertheless, with a wider look at the more recent literature, unusual beliefs in AN may be conceived from a spectrum point of view, as a continuum ranging from good insight (with the possibility for the patient to question his/her conviction) to over-valued ideation and ending with no insight, characterized by delusional thinking (irrefutable beliefs) [11,12,13,53]. Researchers who studied insight in ED populations found very low levels of insight in some patients with restricting AN [14], independent of illness severity [11]. Some authors suggested we should consider thought disorder in AN as a diagnostic specifier for DSM, as in the case of body dysmorphic disorder [54]. Body image distortion in AN appears comparable in the degree of conviction and disruption, but with higher distress than delusional beliefs in schizophrenia [55]. Gadsby (2017) partly ascribes the oversized body experiences in AN to “intuitive certainty” concerning one’s own body schema [56]. There are some recent insights into a possible link between body image disturbances and paranoia in the general population [57]. It is possible that belief inflexibility and failure in integrating disconfirmatory evidence in the thinking process observed in our acute AN patients increases this “intuitive certainty”, exacerbating not only body image disturbances, but also typical anorexic phenomenology, such as rumination, distorted beliefs about eating, food, over-attention to interoceptive signals, misconceptions about energy balance, and also the “eating disorder voice” [7,51]. Focusing on clinical aspects and dividing the sample according to diagnostic subtype, we found that, compared to the restrictive subtype, the binge-eating/purging (ANBP) subtype significantly showed both poorer flexibility in disconfirming their previous judgments (low BADE score) and more willingness to accept absurd interpretations as plausible (high LA score). This result is somewhat surprising since, according to the literature, restricting AN seems to be associated more with thought disorder and greater psychopathology, itself associated with clinical severity [11,14]. Observing the plausibility rates across trials (from the first to the third statement as available information increases), the ANBP group shows a pattern of ratings very close to those reported by psychotic patients [40]; they fail to fully recognize the lure at the end of the task (at stages two and three) and are more prone to accept implausible sentences as plausible, indicating some difficulty in abstract thinking (they accept something that is expressly unlikely as real). 

In agreement with some work on schizophrenia, which found some dependency between belief integration and cognitive functioning [40,58,59], in our patients, biased reasoning processes appeared to be linked to some specific cognitive difficulties (i.e., cognitive rigidity and extreme attention to detail), whereas they appear to be independent of intelligence quotient and age. AN individuals appear unwilling to integrate information that does not support their beliefs or interpretations, especially in the early stages of the process of integrating knowledge (i.e., during the first and the second statements which add information to scenarios), where uncertainty levels are high. This evidence integration bias, as in the case of delusions [38], may contribute to maintaining the high level of rigidity, perfectionism, and obstinacy observed in patients, who continue to make dysfunctional choices (food restriction, hyperactivity, vomiting, …) in spite of adverse consequences. Both from a clinical and neuropsychological point of view, we know how difficult it is for these patients to manage uncertainty [1,11,30,31] and it may be that the ambiguity and uncertainty increase cognitive rigidity and context impermeability. Differently from psychotic literature [24], we have found no link between belief integration bias and clinical characteristics. Besides the strengths and peculiarities of our study, we must note some limitations, which should be addressed by further research. The points to be further investigated and improved concern the lack of comparability of the two samples investigated: in particular, healthy controls have a significantly higher level of education. Although the ANOVA supported our findings, it would be better to have controls with an education level that overlaps that of the patients. Moreover, the sample size of the control group should be increased in order to strengthen and better define our findings. However, our data suggest that there is a cognitive bias not only with respect to the general population but also between diagnostic subtypes, similar to what is observed in the field of psychosis, with respect to the presence or absence of delusional ideation [25]. In line with this, we should extend our data not only to AN subtypes but also to bulimia nervosa, especially in those patients without a prior history of anorexia nervosa [60]. In our study, the sample of patients is mixed, consisting of adolescents and young adults, and it would be better to investigate differentiating adolescents from adults. Other aspects that could be improved are related to patients in clinical remission; in our sample, we investigated only partially remitted patients (i.e., body weight recovered), and with a cross-sectional research design. It would be worth investigating further the symptom remission longitudinally.

To conclude, acute AN people may show reasoning biases in part similar to those observed in psychosis; in the bulimic subtype of AN, thought disorder appeared more prominent and closer to that observed in delusional patients with schizophrenia. Difficulties in flexibility changing individuals’ convictions or beliefs may play a key role not only in developing but also in maintaining eating disorder-related beliefs. Further studies are needed to fully understand the clinical relevance of the presence of reasoning biases in AN and their relationship with insight and response to treatment. A deeper knowledge of the mechanisms underlying eating psychopathology and resistance to treatment in AN would allow us to design tailored interventions to be integrated with existing approaches of proven effectiveness, such as cognitive behavioral therapy, cognitive remediation therapy (CRT), and psychoeducation, similarly to what is already being conducted in schizophrenia with metacognitive treatment [61]. A specific module focused on increasing patients’ awareness of belief integration bias and other thinking traps could be designed and translated into simple exercises to be included within the existing CRT Manual [62].

## Figures and Tables

**Figure 1 jcm-12-01746-f001:**
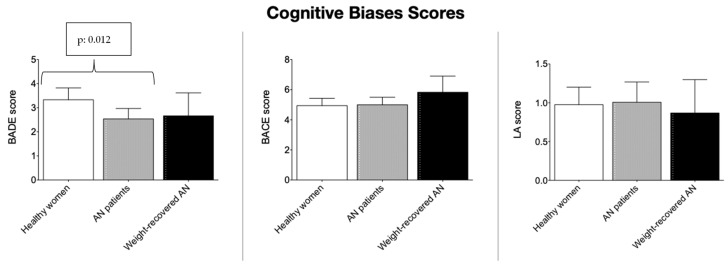
Raw average scores (and their 95% confidence intervals) in the three groups investigated for Bias Against Disconfirmatory Evidence (BADE), Bias Against Confirmatory Evidence (BACE), and liberal acceptance (LA). The error bars indicate the standard error measurement (SEM).

**Figure 2 jcm-12-01746-f002:**
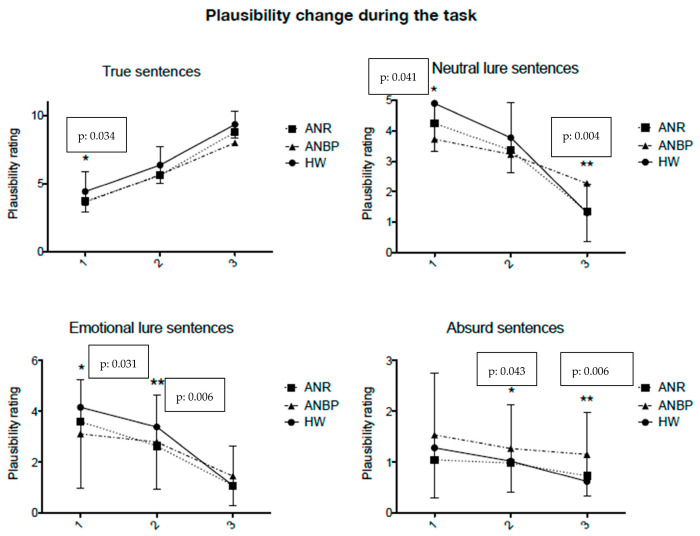
Changes in plausibility rating during the task in healthy women and two clinical groups: ANR and ANBP for the four different types of scenarios. Note: 1, 2, 3: first, second, and third statements progressively adding information to the scenarios as the task progresses; Kruskal–Wallis test; * *p* < 0.05; ** *p* < 0.01. The error bars indicate the SEM.

**Table 1 jcm-12-01746-t001:** Example trial of the BADE task.

	STATEMENTS												
**1**	Jenny can’t fall asleep												
**2**	Jenny can’t wait until it is finally morning												
**3**	Jenny wonders how many presents she will find under the tree												
	**SENTENCES**												
**1**	Jenny is excited about Christmas morning												TRUE
(Implausible)		**0**	**1**	**2**	**3**	**4**	**5**	**6**	**7**	**8**	**9**	**10**	(Very plausible)
**2**	Jenny is nervous about her exam the next day												NEUTRAL LURE
(Implausible)		**0**	**1**	**2**	**3**	**4**	**5**	**6**	**7**	**8**	**9**	**10**	(Very plausible)
**3**	Jenny is worried about her ill mother												EMOTIONAL LURE
(Implausible)		**0**	**1**	**2**	**3**	**4**	**5**	**6**	**7**	**8**	**9**	**10**	(Very plausible)
**4**	Jenny loves her bed												ABSURD
(Implausible)		**0**	**1**	**2**	**3**	**4**	**5**	**6**	**7**	**8**	**9**	**10**	(Very plausible)

Note: Each sentence must be rated with a plausibility score ranging from 0 (implausible) to 10 (very plausible).

**Table 2 jcm-12-01746-t002:** The computation of the dependent variables of the BADE task.

Score	Formula	Meaning
BADE score	[(M_a_EL_1_ + M_a_NL_1_)/2] − [(M_a_EL_3_ + M_a_NL_3_)/2]	Disconfirmatory index
BACE score	|M_a_T_3_ − M_a_T_1_|	Confirmatory index
LA score	[(M_a_A_1_ + M_a_A_2_ + M_a_A_3_)/3]	Absurd index

Note. M_a_EL_1_: the average score given to “emotional lure” interpretations at the first statement; M_a_NL_1_: the average score given to “neutral lure” interpretations at the first statement; M_a_EL_3_: the average score given to “emotional lure” interpretations at the third statement; M_a_NL_3_: the average score given to “neutral lure” interpretations at the third statement; M_a_T_3_: the average score given to “true” interpretations at the third statement; M_a_T_1_: the average score given to “true” interpretations at the first statement; M_a_A_1_: the average score given to “absurd” interpretations at the first statement; M_a_A_2_: the average score given to “absurd” interpretations at the second statement; M_a_A_3_: the average score given to “absurd” interpretations at the third statement.

**Table 3 jcm-12-01746-t003:** Demographic and clinical characteristics of the samples.

	Healthy Women (*n* = 45)	AN Patients (*n* = 82)	Weight-Recovered AN Patients (*n* = 21)
Age	22.8 (4.5)	21.0 (7.1)	21.5 (7.7)
Education (years)	14.5 (2.4) *	12.5 (2.7)	12.1 (2.5)
Body mass index (kg/m^2^)	20.9 (1.8)	15.9 (1.7) ^$^	20.7 (1.8)
Age at onset	…	16.7 (4.6)	15.7 (3.0)
Duration of illness (months)	…	23.6 (36.1)	28.1 (63.1)
Lowest BMI	19.1 (1.8)	14.9 (1.7) ^$^	16.2 (2.1)
Hand lateralization (Edinburgh score)	58.1 (47.1)	57.6 (34.4)	59.5 (36.3)

Note. * significant differences in comparison to the two AN groups; ^$^ significant differences in comparison to both healthy women and weight-recovered patients. All significant comparisons have a *p* value of <0.001, with the exception of the BMI comparison between HC and AN patients, both acute and weight-recovered, which has a *p* value of 0.003.

**Table 4 jcm-12-01746-t004:** Correlations between BADE scores and clinical variables in health women and acute AN.

	Healthy Women (*n* = 45)			AN Patients (*n* = 82)		
	BADE	BACE	LA	BADE	BACE	LA
Age	0.41 **	−0.36 *	0.31 *	0.05	−0.16	0.23 *
Education (years)	0.25	−0.20	0.19	0.38	−0.16	0.21
BMI (kg/m^2^)	−0.20	−0.11	0.29	0.07	0.01	−0.05
Age at onset (years)	…	…	…	0.07	−0.16	0.12
Duration of illness (months)	…	…	…	0.03	0.02	0.16

Note: Spearman’s rho was used as index of correlation; * *p* < 0.05; ** *p* < 0.01.

## Data Availability

The data are available upon request.

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
