# Peer review of "Belief Inflexibility and Cognitive Biases in Anorexia Nervosa—The Role of the Bias against Disconfirmatory Evidence and Its Clinical and Neuropsychological Correlates"

_jcm, 2023, doi:10.3390/jcm12051746_

Round 1
Reviewer 1 Report
I commend the authors for conducting the survey the cognitive biases among anorexia nervosa patients. The rationale is well justified, and data is sound. I only have a few minor concerns.
1. In table 2, the “Education” among “healthy women” and “AN patients” are significantly different, does this affects cognitive biases test? The authors may add a few sentences in discussion.
2. The Discussion part is comprehensive, however, the authors may add a paragraph to discuss the limitations of the current study.
3. In line 113, the “Table 2” appears ahead of “Table 1” in line 178, the authors should sequentially describe them.
4. In Figure 1 and 2, the authors should clearly label whether the error bar indicate SD or SEM.
Reviewer 2 Report
The authors investigated the role/power of cognitive biases in anorexia nervosa (AN), highlighting a few peculiar features of this specific population. Their work is well documented. A few remarks and comments are listed below:
* In the title they could include not just belief inflexibility but a few mental attitudes which were somehow investigated as well. Otherwise, a more generic wording (e.g. mental biases or judgement biases etc. ) could replace the "belief inflexibility" words.
* Abstract: some numerical data should be included with the relative statistical values
* Introduction: please add a few concepts and elaborations concerning the strong influence of early life adversities on the onset and evolution of AN and eating disorders (thus on the patuent's mental attitude etc.)
* Beyond the BADE and BACE biases, you may add few lines and comments on the several possible cognitive biases reported in literature concerning biomedical research and clinical practice (e.g. https://en.wikipedia.org/wiki/List_of_cognitive_biases or https://www.mdpi.com/2077-0383/9/12/4091 )
* Table 2: put the p value to specify that there was/was not any statistically significant difference among the three groups. Please specify that the healthy subject group is under-represented (half in number) in comparison to the AN group, though likely this issue did not alter the statistical analysis etc.
* In figure 1, and where appropriate in other figures, insert the statistical significance (p ?) value in the graphic representation
Comments:
* concerning the investigated subjects' age it could be interesting to examine the bias influence in specific populations, such as the younger subjects who develop AN since the age of 14. Are BADE BACE and LA scores equal in minors in comparison to adults?
* Please make an example of the kind of questions you formulated (to highloght BADE, BACE and LA), in order to clarify to the readers the basic concepts and the compliance of the subjects to the investigation methodology etc.
* In the discussion please elaborate on the practical therapeutic repercussions of this research. Eliciting these biases could lead to a different theraeutic approach and, if so, which ones could be useful ?
